# Thrombin Preconditioning Improves the Therapeutic Efficacy of Mesenchymal Stem Cells in Severe Intraventricular Hemorrhage Induced Neonatal Rats

**DOI:** 10.3390/ijms23084447

**Published:** 2022-04-18

**Authors:** So Yeon Jung, Young Eun Kim, Won Soon Park, So Yoon Ahn, Dong Kyung Sung, Se In Sung, Kyeung Min Joo, Seong Gi Kim, Yun Sil Chang

**Affiliations:** 1Department of Pediatrics, Samsung Medical Center, Sungkyunkwan University School of Medicine, Seoul 06351, Korea; sy0325.jung@sbri.co.kr (S.Y.J.); wonspark@skku.edu (W.S.P.); soyoon.ahn@samsung.com (S.Y.A.); dbible@skku.edu (D.K.S.); sein.sung@samsung.com (S.I.S.); 2Department of Anatomy & Cell Biology, Sungkyunkwan University School of Medicine, Suwon 16419, Korea; kyeungmin.joo@gmail.com; 3Samsung Medical Center, Cell and Gene Therapy Institute, Seoul 06351, Korea; duddms920@skku.edu; 4Department of Health Sciences and Technology, SAIHST, Sungkyunkwan University, Seoul 06351, Korea; 5Center for Neuroscience Imaging Research (CNIR), Institute for Basic Science (IBS), Suwon 16419, Korea; seonggikim@skku.edu; 6Department of Biomedical Engineering, Sungkyunkwan University, Suwon 16419, Korea

**Keywords:** mesenchymal stem cell transplantation, cerebral intraventricular hemorrhages, translational medical research, infant, newborn, diseases

## Abstract

Severe intraventricular hemorrhage (IVH) remains a major cause of high mortality and morbidity in extremely preterm infants. Mesenchymal stem cell (MSC) transplantation is a possible therapeutic option, and development of therapeutics with enhanced efficacy is necessary. This study investigated whether thrombin preconditioning improves the therapeutic efficacy of human Wharton’s jelly-derived MSC transplantation for severe neonatal IVH, using a rat model. Severe neonatal IVH was induced by injecting 150 μL blood into each lateral ventricle on postnatal day (P) 4 in Sprague-Dawley rats. After 2 days (P6), naïve MSCs or thrombin-preconditioned MSCs (1 × 10^5^/10 μL) were transplanted intraventricularly. After behavioral tests, brain tissues and cerebrospinal fluid of P35 rats were obtained for histological and biochemical analyses, respectively. Thrombin-preconditioned MSC transplantation significantly reduced IVH-induced ventricular dilatation on in vivo magnetic resonance imaging, which was coincident with attenuations of reactive gliosis, cell death, and the number of activated microglia and levels of inflammatory cytokines after IVH induction, compared to naïve MSC transplantation. In the behavioral tests, the sensorimotor and memory functions significantly improved after transplantation of thrombin-preconditioned MSCs, compared to naïve MSCs. Overall, thrombin preconditioning significantly improves the therapeutic potential and more effectively attenuates brain injury, including progressive ventricular dilatation, gliosis, cell death, inflammation, and neurobehavioral functional impairment, in newborn rats with induced severe IVH than does naïve MSC transplantation.

## 1. Introduction

Intraventricular hemorrhage (IVH), caused by hemorrhage in the friable germinal matrix vasculature, is a serious complication in extremely premature infants. Ensuing brain damage due to severe IVH and subsequent progression to post-hemorrhagic hydrocephalus (PHH) increase mortality and severe morbidities, such as seizures, cerebral palsy, and neurodevelopmental delay in preterm infants [1,2]. In the center of pathogenesis of IVH-induced brain injury in preterm infants, inflammatory responses and apoptotic cell death caused by blood entering the ventricle occur, which results in white matter injury and cortical neuronal damage; subsequently progressive hydrocephalus develops, which further exacerbates pre-existing brain injuries [3]. However, despite recent advances in perinatal medicine, there are no clinically effective treatments to attenuate brain damage and prevent progression to PHH in premature infants with severe IVH, Therefore, a new treatment strategy is urgently needed.

We have previously demonstrated that transplantation of human umbilical cord blood (UCB)-derived mesenchymal stem cells (MSCs) into neonatal rats with IVH effectively attenuated brain injury and PHH through anti-inflammatory and anti-apoptotic paracrine effects, rather than regeneration [3,4]. We have also shown that brain-derived neurotrophic factor (BDNF) secreted from transplanted MSCs is one of the important factors that play a pivotal role in attenuating IVH-induced brain injuries in newborn rats [5] and attenuates hippocampal neuronal loss and circuit damage after IVH induction through BDNF-CREB signaling [6]. Furthermore, we have shown that safety and feasibility of transplantation of allogenic MSCs in phase 1 clinical trial in extremely preterm infants with grade 4 IVH [2]. These data suggest that MSC therapy is a promising option for treating IVH in premature infants.

Although naïve MSCs themselves yield significant therapeutic effects, a novel strategy is needed to maximize the therapeutic effect of MSCs for severe diseases, such as high- grade IVH-induced brain injury. Increasing evidence has shown that preconditioning of MSCs, including exposure to hypoxia, lipopolysaccharides, growth factors, hormones, and pharmacological or chemical agents, can optimize the paracrine potency and therapeutic efficacy of MSCs [7,8,9]. Previously, we have shown that among various preconditioning conditions, thrombin is superior to hypoxia, lipopolysaccharide, or H_2_O_2_ upon biosynthesis and enrichment of cargo contents of exosomes derived from UCB-derived MSCs [10,11]. From the perspective of industrial development, we then used Wharton’s jelly (WJ)-derived MSCs because of their possible advantage for industrial use over UCB-derived MSCs, considering that they are easier to harvest, expand, and produce on a large scale [12]. Thrombin preconditioning has been shown to enhance the efficacy of WJ-derived MSCs in vitro and in vivo in a neonatal hypoxic–ischemic brain injury [13]. Therefore, in this study, we aimed to evaluate whether thrombin preconditioning of human WJ-derived MSCs can improve the anti-apoptotic, anti-inflammatory, and neuroprotective effects of naïve MSCs, thereby more effectively attenuating severe IVH-induced brain injury than naïve MSCs, using a severe IVH model of newborn rats with significant brain injury.

## 2. Results

### 2.1. Cell Viability, Cytotoxicity, Oxidative Stress and Cell Death Assay

Thrombin-induced neuronal cell injury, an in vitro model of IVH, significantly reduced cell viability and increased cytotoxicity as evidenced by increased levels of lactate dehydrogenase (LDH) and oxidative stress evidenced by increased level of malondialdehyde (MDA), respectively, compared to normal control (Figure 1A–C). However, treatment with naïve MSCs significantly improved cell viability and reduced the LDH and MDA levels. Thrombin preconditioning of MSCs significantly enhanced the neuronal protective effects of naïve MSCs on cell viability, cytotoxicity, and oxidative stress. In line with this, thrombin-induced increase in the number of terminal deoxynucleotidyl transferase dUTP nick end labeling (TUNEL)-positive cells was significantly reduced after treatment with naïve MSCs; further, the protective effect of naïve MSCs was significantly enhanced in thrombin-preconditioning of MSCs (Figure 1D). In the comparison of the ability to secrete BDNF between naive and thrombin-preconditioned MSCs, the secretion level of BDNF was significantly higher after thrombin preconditioning than in after naïve MSCs transplantation (Figure 1E).

### 2.2. Survival and Body Weight

Neonatal IVH was induced at postnatal day (P4) by infusing 150μL of fresh whole blood obtained from the mother rat into the bilateral ventricles respectively. Two days after IVH induction, naïve or thrombin preconditioned MSCs were transplanted into the ventricle. Survival rate and body weight was observed from P4 to P35. The survival rates were significantly reduced in the IVH control (IC) and IVH treatment with naïve MSCs (INM) groups but not significantly different between the IVH treatment with thrombin-preconditioned MSCs (ITM) and normal control (NC) groups. The survival rates in the NC, IC, INM and ITM groups were 100%, 80%, 74%, and 88%, respectively (Appendix A). Weight gain was significantly reduced at 4 weeks after IVH induction in all IVH-induced groups compared to that in the NC group, but was not statistically different between the IC, INM, and ITM groups. The final body weights in the NC, IC, INM, and ITM groups were 145 ± 3 g, 130 ± 3 g, 131 ± 3 g, 131 ± 3 g, respectively (Appendix A).

### 2.3. Serial Brain Magnetic Resonance Imaging (MRI) Analysis

Ventricular dilatation in the IVH-induced group was confirmed and measured using brain MRI on postnatal day (P) 5, P11, and P33, as shown in Figure 2A. One day after IVH induction (at P5), the ventricular volume was not significantly different among IC, INM and ITM groups. However, a progression of ventricular dilatation following posthemorrhagic hydrocephalus at P10 and P33 was clearly observed in IC group. Significant reduction in ventricle volume was observed in ITM group but not in INM group. Also, there was no significant difference in the ventricular volume of the INM group compared to the IC group at P33 (Figure 2B).

### 2.4. Reactive Gliosis and Cell Death

Reactive gliosis and cell death are universal reactions to inflammatory brain injury. A remarkable increase in the fluorescence intensity of glial fibrillary acidic protein (GFAP), a marker of reactive gliosis, was observed in all IVH-induced groups compared to that in the NC group (Figure 3A). However, the increased fluorescence intensity of GFAP in the IC group was significantly reduced after transplantation of thrombin-preconditioned MSCs but not after transplantation of naïve MSCs.

A marked increase in the number of TUNEL-positive cells was observed in all IVH-induced groups compared to that in the NC group (Figure 3B). The increased number of TUNEL-positive cells in the IC group was significantly reduced after transplantation of naïve and thrombin-preconditioned MSCs. However, the reduction in the number of TUNEL-positive cells are higher in the ITM group than in the INM group.

### 2.5. Brain Inflammation

Activated form of microglia is observed at the site of brain inflammation. The number of ED1-positive cells, a marker of activated form of microglia, was significantly higher in all IVH-induced groups than in the NC group (Figure 3C). The increase in the number of ED1-positive cells was significantly reduced after transplantation of thrombin-preconditioned and naïve MSCs compared to IC group; however, the reduction in the number of ED1-positive cells was higher in the ITM group than in the INM group. In line with this, the significantly increased levels of inflammatory cytokines, such as interleukin (IL)-1α, IL-1β, IL-6, and tumor necrosis factor (TNF)-α, in the IC group compared to those in the NC group were significantly reduced after transplantation of thrombin-preconditioned and naïve MSCs (Figure 4). The reduction in these inflammatory cytokine levels was more significant after transplantation of thrombin-preconditioned MSCs than after transplantation of naïve MSCs.

### 2.6. Behavioral Function Test

To evaluate the sensorimotor and memory functions of the IVH-induced rats, we performed the negative geotaxis, rotarod and passive avoidance tests. In the negative geotaxis test at P32, only the IC group showed a significant delay in the reaction time for reorienting themselves compared to the NC group (Figure 5A). The results slightly improved in the negative geotaxis test after transplantation of naïve and thrombin-preconditioned MSCs; however, the differences were not significant compared to IC group (NC, IC, INM, and ITM group: 3.2 ± 0.1, 4.2 ± 0.2, 3.9 ± 0.2, and 3.7 ± 0.2, respectively). In the rotarod test performed form P31 to P33, all IVH-induced groups showed a decrease in the motor function compared to the NC group. A significant impairment in the motor function was observed in the IC and INM groups, but not in the ITM group, compared to that in the NC group. After transplantation of thrombin-preconditioned MSCs, a significant improvement in the ITM group was observed compared to that in the IC group at P33 (NC, IC, INM, and ITM group: 221 ± 12.09, 141 ± 12.1, 160 ± 16.0, and 192 ± 15.0, respectively) (Figure 5B). In line with this, the latency (s) to enter the dark chamber in the passive avoidance test in the IC and IM groups were significantly reduced compared to that in the NC group. However, the reduced latency significantly improved in the ITM group compared with that in the IC group (NC, IC, INM, and ITM group: 147 ± 8.5, 56 ± 8.9, 87 ± 12.7, and 109 ± 8.6, respectively) (Figure 5C).

## 3. Discussion

Despite advances in neonatal intensive care, IVH and its side effects are still associated with high mortality and long-term morbidity, as no effective treatment exists. In our previous studies, we have demonstrated that MSCs have cell protective and anti-inflammatory effects on neonatal brain injuries, such as IVH, hypoxic–ischemic encephalopathy (HIE), and meningitis [4,14,15]. However, a novel strategy for enhancing the therapeutic effect of MSCs is necessary to maximize their therapeutic potential including paracrine potency, immediately after transplantation, especially for severe injuries, such as severe IVH, in premature infants. Herein, we hypothesized that the efficacy of naïve MSCs could be enhanced by pre-exposure to thrombin, which is increased in hemorrhagic and ischemic brain injuries [16], in a preclinical model of severe IVH. In present study, we observed that thrombin-preconditioned MSCs significantly enhanced the anti-apoptotic, anti-cytotoxic, and antioxidant effects MSCs in an in vitro model of thrombin-induced neuronal injury, and anti-apoptosis, anti-gliosis and anti-inflammation in an in vivo model of severe IVH in newborn rats. In brain MRI follow-up and neurobehavioral studies, thrombin-preconditioned MSCs, but not naïve MSCs, showed a significant effect on neuroprotection, such as attenuation of PHH and improvement of memory function.

Developing an appropriate animal model is essential for successful clinical application. In severe IVH, high grades of IVH (grade 3 and 4), blood is found in the brain ventricles. Blood clots can block the flow of cerebrospinal fluid and lead to hydrocephalus, which greatly increases mortality (approximately 20–30% of mortality) and permanently impairs physical and mental developments, compared with no or low-grade of IVH [17]. In this study, we developed a preclinical model of severe IVH in newborn rats on P4. To simulate the clinical condition of IVH, we injected maternal whole blood into the brain ventricle of rat pups on P4, which is comparable to 27 weeks of gestational age in human [18]. Our analysis revealed significantly increased mortality, progressive PHH, cell death, inflammation, reactive gliosis, and impaired behavioral function, indicating that the animal model reflected the clinical pathology of severe neonatal IVH for testing the efficacy of new treatment for severe neonatal IVH. We intraventricularly injected 1 × 10^5^ of MSCs into newborn rats on P6. The feasibility of the MSC injection method was determined on the basis of the pre-clinical dose, route, and timing of MSC transplantation [4,19,20].

MSCs can be preconditioned in various methods, such as pre-stimulation with hypoxia, lipopolysaccharide, H_2_O_2_, and thrombin. In our previous report, thrombin preconditioning of MSCs, compared with other preconditioning regimens that involves exposure to hypoxia, lipopolysaccharide and H_2_O_2_, showed stronger effect on in vivo wound healing capacity as well as cell proliferation and anti-oxidant and anti-apoptotic processes, boosting the production of extracellular vesicles, which are paracrine mediators of MSCs [10], via protease-activated receptor-mediated signaling [11]. Moreover, thrombin preconditioning of MSCs remarkably enriched their cargo contents, such as angiogenin, hepatocyte growth factor, vascular endothelial growth factor and BDNF, which are known to alleviate brain damage after stroke [21], compared to naïve MSCs [10,13]. We have previously observed that the mRNA and protein levels of BDNF in UCB-derived MSCs were upregulated after exposure to thrombin, and that BDNF was a critical paracrine mediator of the therapeutic efficacy of MSCs for neonatal IVH-induced brain injury, attenuating cell death, astrogliosis, inflammatory response, and PHH and improving neurogenesis and myelination [5]. In our previous study, thrombin-preconditioned MSCs secreted higher level of BDNF and showed a more significant therapeutic effect for neonatal HIE-induced brain injury than did naïve MSCs [13]. In this study, we confirmed that the paracrine capacity of MSCs to secrete trophic factors, such as BDNF, significantly increased after thrombin preconditioning compared to that after the naïve control treatment. These results suggest that thrombin preconditioning improves the paracrine capacity of MSCs, thereby meaningfully enhancing the therapeutic efficacy of naïve MSCs, which might not be sufficient in in vivo models, even though it is meaningful in in vitro IVH model. However, in the present study, since we have not directly compared the efficacy of thrombin preconditioned MSCs and EVs isolated from thrombin-preconditioned MSCs, it is unknown whether the efficacy of these two conditions would be similar or not. Thus, further studies are needed to clarify this.

We speculated several reasons why the therapeutic potency of naïve MSCs appeared to be meaningful in in vitro studies but not as well as that of thrombin-preconditioned MSCs in in vivo studies. First, our study was conducted using a more severe IVH in vivo model. In a previous study, brain injury was induced via infusion of 200 μL of maternal blood to the brain ventricle [2,3,4,5,20]. In this study, we injected 300 μL of maternal blood to develop a more severe IVH model. The neuroprotective effect of naïve MSCs might not be sufficient to attenuate high-grade IVH, and the inordinately harsh condition of the brain with severe IVH might inhibit naive MSCs from properly activating to survive and secreting trophic factors. Second, cell physiological characteristics, such as the proliferation capacity and paracrine potency of MSCs, could differ depending on the tissue sources, such as UCB and WJ [22] as well as on the lot variation from even the same source [23]. Thus, further studies are needed for clarification.

In summary, our study in a severe IVH neonatal model showed that thrombin preconditioning significantly enhanced the efficacy of MSCs, attenuating cytotoxicity, cell death, PHH, and inflammation and improving behavioral functions. These findings suggest that thrombin preconditioning may be a feasible option for maximizing the efficacy of MSCs for severe IVH-induced brain injury.

## 4. Materials and Methods

### 4.1. Mesenchymal Stem Cell Preparation

After obtaining informed consent from pregnant mothers, we isolated, expand, and preconditioned human WJ-derived MSCs with thrombin (2 U/mL; Sigma Aldrich, Steinheim, Germany) at Good Manufacturing Practice (GMP) facility of the Samsung Stem Cells and Regenerative Medicine Institute following the guidelines for GMP, as described previously [13]. Briefly, the MSCs characteristics were confirmed using flow-cytometric analysis for cell surface markers (CD73, CD90, CD105, CD166, CD14, CD11b, HLA-DR (MHCII), CD34, CD45, and CD19) and in vitro differentiation assays for osteogenesis, adipogenesis, and chondrogenesis as described [24]. After reaching 90% confluence, we preconditioned WJ-derived MSCs with thrombin in a culture medium (α-MEM; Gibco, Life Technologies, Carlsbad, CA, USA) for 3 h. Control naïve MSCs were prepared in the same manner except for thrombin treatment.

### 4.2. In Vitro Model of Thrombin-Induced Neuronal Injury

Cerebral cortical neuronal cells were primarily cultured, as described previously [5]. Briefly, the neuronal cells were exposed to thrombin (40U; Reyon Pharmaceutical Co., Ltd., Seoul, Korea) overnight to mimic in vivo hemorrhagic conditions. After thrombin-induced neuronal cell injury, naïve MSCs or thrombin-preconditioned MSCs were treated with a concentration of 1 × 10^5^ cells per 1 mL. The neuronal cells were harvested and analyzed after 24 h of co-incubation with MSCs.

### 4.3. In Vitro Cell Viability, Cytotoxicity, Oxidative Stress and Cell Death Assays

Cell viability was evaluated using a cell counting kit-8 (Dojindo, Kumamoto, Japan) assay (Dojindo Molecular Technologies Inc., Rockville, MD, USA), according to the manufacturer’s instructions. The relative cell viability (%) was determined by normalizing to 100% (untreated cells) controls. Cytotoxicity was determined using LDH assay kit (Roche, Mannheim, Germany), according to the manufacturer’s instructions. Oxidative stress was measured using Oxiselect TBARS assay kit (MDA Quantitation) (Cell Biolabs, San Diego, CA, USA), according to the manufacturer’s instructions. The number of dead cells was counted in un-overlapping 5-high power fields using TUNEL assay.

### 4.4. Animal Model

All animal experiments were reviewed and approved by Institutional Review Board of Sungkyunkwan University. This study followed the institutional and National Institutes of Health guidelines for laboratory animal care. All animal procedures were performed in an AAALAC-accredited specific pathogen-free facility at Sungkyunkwan University. Newborn Sprague-Dawley rats were purchased from Orient Co (Seoul, Korea). The pups were kept with a nursing mother rat for the first 2 weeks of life and then housed in individual cages with free access to laboratory chow and water thereafter. To control for gender differences in brain damage, we selectively used male rat pups [25]. The experiment started at postnatal day 4 (P4) and ended at P35. To induce more severe IVH than that in our previous study [3,4,5], the P4 rat pups were anesthetized with 1.5–3.0% isoflurane in oxygen-enriched air and injected at a rate of 80 μL/min with 300 μL of fresh whole blood obtained from the caudal vein of the mother rat into the bilateral ventricles (150 μL into each ventricle) under stereotaxic guidance (Digital Stereotaxic Instrument with Fine Drive MyNeurolab, St. Louis, MO, USA; coordinates relative to the bregma: x = ±0.5, y = −0.5, z = +2.5 mm) [4,26]. One day after, the severe IVH induction was confirmed at P5 using T2-weighted anatomical images obtained using 9.4-Tesla Bruker BioSpec MRI system (Billerica, MA, USA). The rat pups were randomly divided into the IVH control group (IC, *n* = 29), an IVH treatment with naïve MSCs group (INM, *n* = 23), and an IVH treatment with thrombin-preconditioned MSCs group (ITM, *n* = 28). Two days after IVH induction, naïve or thrombin-preconditioned MSCs (1 × 10^5^ cells in 10 μL saline) were slowly injected at very low rate (10 μL/min) into the right ventricle under stereotaxic guidance (Digital Stereotaxic Instrument with Fine Drive MyNeurolab; coordinates relative to the bregma: x = +0.5, y = −0.5, z = +2.5 mm) [19]. An equal volume of saline was administered to the rats in the control group. Figure 6 shows the experimental schedules in detail.

### 4.5. Serial Brain MRI and Ventricle-to-Whole Brain Ratio assessment

After IVH induction at P4, approximately 15–20% of the initial brain damage was confirmed at P5 and at a follow-up at P10 and P33, using the 9.4 Tesla MRI system. During MRI, the rat pups were monitored under 2% isoflurane in oxygen-enriched air. Twelve coronal images were obtained at an in-plane resolution of 0.1 × 0.1 mm^2^ and a slice thickness of 1.0 mm without inter-slice gap. On average, one MRI session took 30 min per pup. The ventricle-to-whole brain volume ratio was calculated to monitor the progress of PHH, as described previously [4]. The degree of ventricular dilatation in each experimental animal was measured using the ImageJ program (National Institutes of Health, Bethesda MD, USA).

### 4.6. Immunohistochemical Analysis

To histologically evaluate brain injury, we coronally paraffin-sectioned brain tissues at 5 μm after 24-h fixation with 4% paraformaldehyde. Reactive gliosis and brain inflammation were evaluated by immunostaining for GFAP and ED1. Brain tissues were stained with GFAP antibody (1:1000; IR527 A11001, Dako, Glostrup, Denmark) and ED1 antibody (1:250; ab31630, Abcam, Cambridge, MA, USA). GFAP- and ED1-positive cells were visualized with Alexa 568 secondary antibodies and photographed under a microscope at 200× magnification. The fluorescence intensity of GFAP was measured using ImageJ. The number of ED1-positive cells was counted in high power fields. The fluorescence intensity of GFAP and number of ED-1-positive cells were measured on six non-overlapping fields in the coronally sectioned brains (+0.95 to −0.11 mm/bregma) in the periventricular zone; two fields (right and left ventricles) were captured in one brain slice and three brain slices were analyzed. The immunostained brain area was counterstained with 4′,6-diamidine-2′-phenylindole dihydrochloride (DAPI; Roche, Basel, Switzerland) to visualize the nuclei. Each stained brain tissue was scanned using a confocal laser scanning microscope (LSM 700, Zeiss, Oberkochen, Germany) by blinded observers.

### 4.7. TUNEL Assay

Cell death-associated DNA fragmentation, a marker of dead cells, was identified via TUNEL staining, according to the manufacturer’s instructions for the DeadEnd Fluorometric TUNEL System kit (G3250; Promega, Madison, WI, USA) and counterstained with DAPI. After scanning by blinded observers using the confocal laser scanning microscope, the number of TUNEL-positive cells was counted in six non-overlapping fields in the coronally sectioned brains (+0.95 to −0.11 mm/bregma) in the periventricular zone; two fields (right and left ventricle) were captured in one brain slice and three brain slices were analyzed.

### 4.8. Enzyme-Linked Immunosorbent Assay (ELISA)

The levels of inflammatory cytokines, such as IL-1α, IL-1β, IL-6 and TNF-α, in the cerebrospinal fluid were measured using commercial ELISA kits (R&D systems, Minneapolis, MN, USA), according to the manufacturer’s instructions. The expression levels of BDNF were measured using a BDNF ELISA kit (R&D system), according to the manufacturer’s instructions.

### 4.9. Behavioral Function Test

The negative geotaxis test was performed to evaluate reflex motor coordination and vestibular sensitivity in the young rodent model [27]. The rats were placed head-down on an inclined wooden platform and then gently held for 3 s in this position. After release, the time required for the rats to reorient themselves (180° rotation) to face the uphill was recorded. The recorded time was averaged in triplication, at P11, P18, P25 and P32 [28]. 

The rotarod test was used to evaluate motor skill learning and neuromuscular coordination [29] using a rotarod apparatus (Ugo Basile, Comerio, Italy). The time for which the rats were able to remain on the rod turning at increasing speed was recorded for 3 consecutive days, from P31 to P33, considering their ability to learn rotarod performance. The test was performed three times a day, with 15 min rest intervals, and the results were averaged [30,31].

The passive avoidance test was performed to evaluate the learning and memory of the rats [32]. The passive avoidance apparatus consisted of two compartments (dark and light chambers) that were separated from each other by a guillotine door. In the preliminary test, after the initial 3 min of acclimatization period, the rats were placed in the light chamber and allowed to enter the dark chamber. Immediately after the rats entered the dark chamber, the guillotine door was closed, and an electric foot shock stress (0.5 mA) was delivered for 10 s. The rats were left for 10 s to recognize the shock stress in the dark chamber. The main test was performed 24 h after the preliminary test. Following the acclimatization period, the rats were exposed to an electric foot shock (0.5 mA), and the latency time to the dark chamber was recorded. This performance was observed for up to 180 s. When the animal remained in the light chamber without stepping down to the dark chamber, the latency time was recorded as 180 s. Each evaluation was conducted independently with blinding of the experimental groups [6].

### 4.10. Statistical Analysis

Data were presented as means ± standard errors of the mean. Statistical comparisons between the groups were performed using one-way analysis of variance and Tukey’s post hoc analysis. All data were analyzed using the SAS software (version 9.4; SAS Institute, Cary, NC, USA). Statistical significance was set at *p* < 0.05.

## Figures and Tables

**Figure 1 ijms-23-04447-f001:**
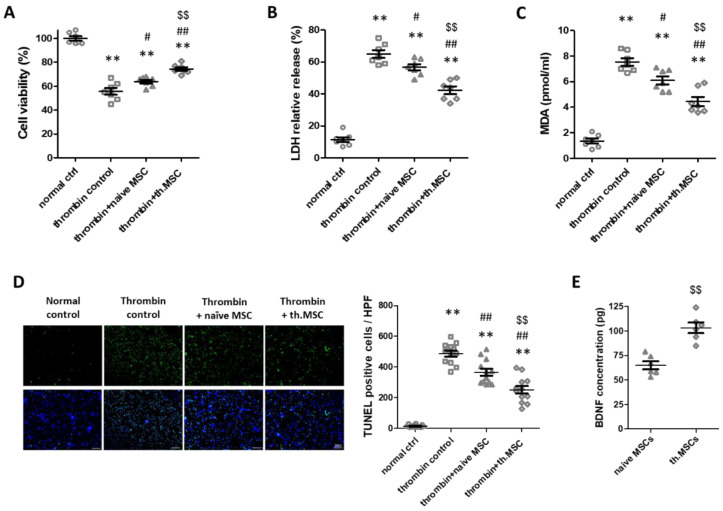
Effect of thrombin preconditioning on the neuroprotective efficacy of MSCs in the in vitro thrombin-induced neuronal injury model. (**A**–**C**) Cell viability, expressed as the relative proliferation rate (%) to the normal control, cytotoxicity, expressed as the relative LDH level, MDA level, (*n* = 7/group) and (**D**) the number of TUNEL-positive cells captured using florescent microscopy (green: original magnification; ×400, scale bars: 20 μm) evaluated in the rat primary cultured cortical neurons 24 h after co-culture with naïve or thrombin-preconditioned MSCs (*n* = 12/group). The enlarged images are shown in Appendix A (**E**) BDNF protein levels measured in culture medium of naïve MSCs and thrombin-preconditioned MSCs. All the in vitro analyses were performed using MSCs prepared at one time (*n* = 6/group). Data are expressed as means ± SEM. ** *p* < 0.01 compared to the normal control group, ^#^
*p* < 0.05 and ^##^
*p* < 0.01 compared to thrombin induction control group, ^$$^
*p* < 0.01 compared to thrombin induction with naïve MSC treatment group. BDNF, brain-derived neurotrophic factor; LDH, lactate dehydrogenase; MDA, malondialdehyde; MSCs, mesenchymal stem cells; SEM, standard error of the mean; TUNEL, terminal deoxynucleotidyl transferase dUTP nick end labeling.

**Figure 2 ijms-23-04447-f002:**
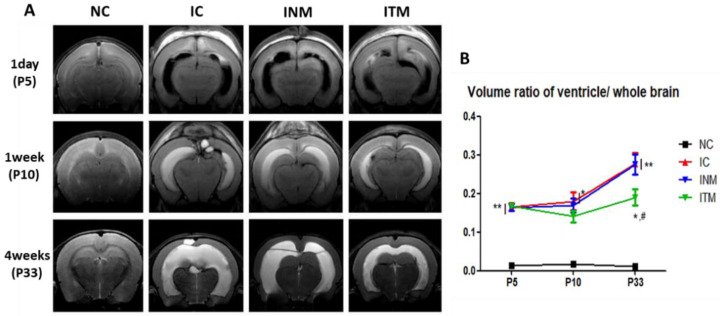
Reduced ventricular dilatation induced by severe IVH observed after transplantation of thrombin- preconditioned MSCs. Transplantation of thrombin-preconditioned MSCs reduced ventricular dilatation after IVH induction. (**A**) Serial brain magnetic resonance images of the four groups and (**B**) ventricle-to-whole brain volume ratio (*n* = 18, 29, 23, 28 in the NC group, IC group, INM group and ITM group, respectively). Data are expressed as means ± SEM. * *p* < 0.05 and ** *p* < 0.01 compared to the NC group, ^#^
*p* < 0.05 compared to the IC group. NC, normal control; IC, IVH control; INM, IVH with transplantation of naïve MSCs; ITM, IVH with transplantation of thrombin-preconditioned MSCs; IVH, intraventricular hemorrhage; MSCs, mesenchymal stem cells; SEM, standard error of the mean.

**Figure 3 ijms-23-04447-f003:**
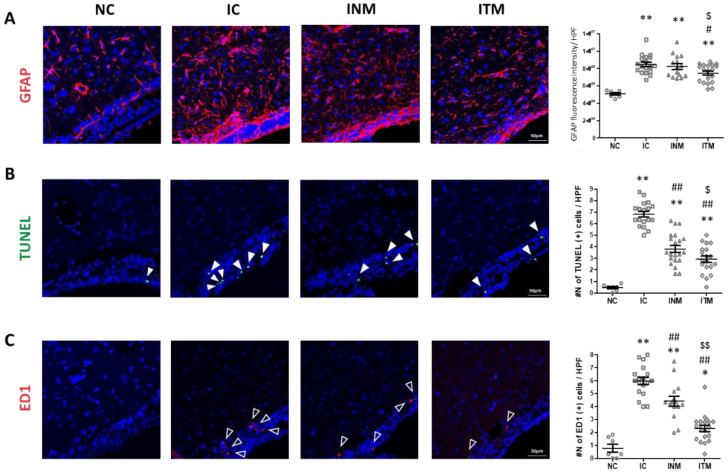
Decreases in reactive gliosis, activated microglia, and cell death induced by severe IVH after treatment with thrombin-preconditioned MSCs. Immunofluorescence micrographs and quantified averages of the ventricular area with (**A**) GFAP (red), (**B**) TUNEL (green), and (**C**) ED-1 (red) staining. The nucleus was visualized with DAPI (blue) (original magnification, ×200). (*n* = 7, 18, 15 and 19 in the NC group, IC group, INM group and ITM group, respectively.) Data are expressed as means ± SEM. * *p* < 0.05 and ** *p* < 0.01 compared to the NC group, ^#^
*p* < 0.05 and ^##^
*p* < 0.01 compared to the IC group, ^$^
*p* < 0.05 and ^$$^
*p* < 0.01 compared to the treatment with naïve MSCs group; NC, normal control; IC, IVH control; INM, IVH with transplantation of naïve MSCs; ITM, IVH with transplantation of thrombin-preconditioned MSCs; IVH, intraventricular hemorrhage; MSCs, mesenchymal stem cells; GFAP, glial fibrillary acidic protein; TUNEL, terminal deoxynucleotidyl transferase dUTP nick end labeling; DAPI, 4′,6-diamidine-2′-phenylindole dihydrochloride; SEM, standard error of the mean.

**Figure 4 ijms-23-04447-f004:**
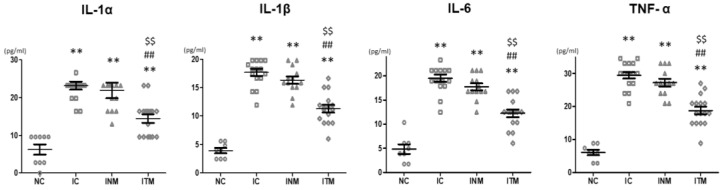
Reduced levels of inflammatory cytokines (IL-1α, IL-1β, IL-6 and TNF-α) in the CSF observed in the rats with severe IVH after transplantation with thrombin-preconditioned MSCs (*n* = 8, 16, 13 and 16 in the NC group, IC group, INM group and ITM group, respectively). Data are expressed as means ± SEM. ** *p* < 0.01 compared with the NC group, ^##^
*p* < 0.01 compared with the IC group, ^$$^
*p* < 0.01 compared with the treatment with naïve Wharton’s jelly-derived MSCs group. NC, normal control; IC, IVH control; INM, IVH with transplantation of naïve MSCs; ITM, IVH with transplantation of thrombin-preconditioned MSCs; IL, interleukin; TNF, tumor necrosis factor; CSF, cerebrospinal fluid; IVH, intraventricular hemorrhage; MSCs, mesenchymal stem cells; SEM, standard error of the mean.

**Figure 5 ijms-23-04447-f005:**
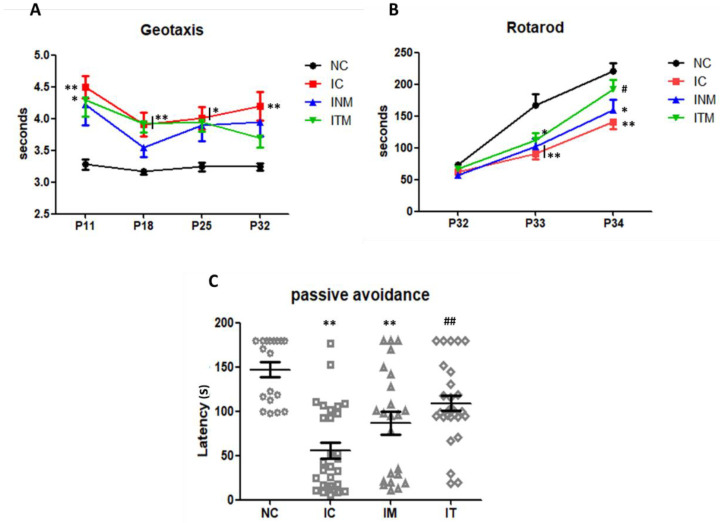
Improvement of the sensorimotor and memory functions in the rats with severe IVH after transplantation with thrombin-preconditioned MSCs. (**A**) Neurobehavioral functional outcomes in the negative geotaxis test, (**B**) rotarod test, and (**C**) passive avoidance test. (*n* = 18, 29, 23, 28 in the NC group, IC group, INM group and ITM group, respectively) Data are expressed as means ± SEM. * *p* < 0.05 and ** *p* < 0.01 compared with the NC group, ^#^
*p* < 0.05 and ^##^
*p* < 0.01 compared with the IC group. P, postnatal day; NC, normal control; IC, IVH control; INM, IVH with transplantation of naïve MSCs; ITM, IVH with transplantation of thrombin-preconditioned MSCs; IVH, intraventricular hemorrhage; MSCs, mesenchymal stem cells; SEM, standard error of the mean.

**Figure 6 ijms-23-04447-f006:**
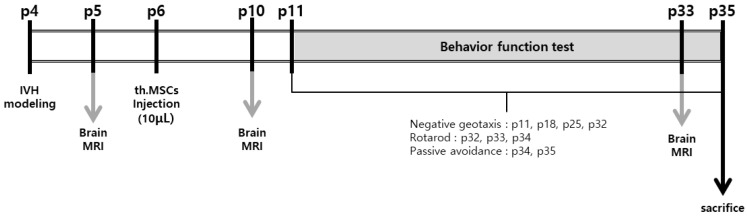
IVH experimental protocol. All injections were performed directly into the lateral ventricle. IVH, intraventricular hemorrhage; MRI, magnetic resonance imaging; MSCs, mesenchymal stem cells; th.MSCs, thrombin-preconditioned mesenchymal stem cell.

## Data Availability

Not applicable.

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
