# Peer review of "Thrombin Preconditioning Improves the Therapeutic Efficacy of Mesenchymal Stem Cells in Severe Intraventricular Hemorrhage Induced Neonatal Rats"

_ijms, 2022, doi:10.3390/ijms23084447_

Round 1

Reviewer 1 Report

Jung et al. demonstrated the efficacy of thrombin preconditioned MSCs, in comparison to naïve MSCs, in treating severe intraventricular hemorrhage in neonatal rats. In vitro, preconditioning of MSCs promoted neuronal cell survival and reduced cytotoxicity. In vivo, thrombin preconditioned MSCs reduced cell death and release of inflammatory cytokines. Rats treated with preconditioned MSCs also showed improved functional behaviour compared with Rats treated with naïve MSCs. This study has demonstrated the potential therapeutic use of MSCs for treating IVH. 

  1. The provided reference (#12) supports the use of Wharton’s jelly-derived MSCs and umbilical cord blood (UCB)-derived MSCs as a source of diabetes mellitus cell therapy, in term of insulin production. Is there any data support/ compare the use of Wharton’s jelly- and UCB-derived MSCs in study similar to the current study? Such as neuronal cytotoxicity or inflammation?
  2. Some of the terms or reasoning for the experiments are not clear for readers. For examples, What is the function of glial fibrillary acidic protein and ED-1? How is it related to cell death?
  3. The authors have published a series of interesting studies regarding thrombin-preconditioning of MSCs. It seems like the beneficial effects of thrombin preconditioned MSCs is due to the increase production of extracellular vesicles (EVs). How is the transplantation of preconditioned MSCs compared to EVs secreted from the preconditioned MSCs in treating IVH in vivo?
  4. In line 224, the author mentioned about the antioxidant effects of MSCs, it is not clear which data supported this statement.
  5. In Method section, Oxiselect TBARS assay is mentioned. Where is the result?
  6. The levels of secretory inflammatory cytokines were measured from the cerebrospinal fluid. Is the any histological assessment of these cytokines in the brain tissues?

Author Response

  • Reviewer #1

1-1) The reviewer stated, “The provided reference (#12) supports the use of Wharton’s jelly-derived MSCs and umbilical cord blood (UCB)-derived MSCs as a source of diabetes mellitus cell therapy, in term of insulin production. Is there any data support/ compare the use of Wharton’s jelly- and UCB-derived MSCs in study similar to the current study? Such as neuronal cytotoxicity or inflammation?”

-> We appreciate the reviewer’s comments. Although we previously conducted an experiment using UCB-derived MSCs in the Intraventricular hemorrhage (IVH) neonatal rat model (reference #4), we used Wharton’s jelly (WJ)-derived MSCs in the present study considering the perspective of industrial development, since WJ-derived MSCs are easier to harvest, expand, and produce on a larger scale than UCB-derived MSCs (lines 72-75), as mentioned in reference #12. We do not have any data on comparison between the efficacy of UCB- and WJ-derived MSCs for neuronal cytotoxicity or inflammation occurring in neonatal IVH animal models.

1-2) The reviewer stated, “Some of the terms or reasoning for the experiments are not clear for readers. For examples, what is the function of glial fibrillary acidic protein and ED-1? How is it related to cell death?”

-> Histologically, we observed levels of glial fibrillary acidic protein (GFAP), TUNEL-positive cells, and ED-1-positive cells as markers for reactive gliosis, apoptotic cell death, and activated microglia, respectively, because increases in reactive gliosis, cell death, and microglia activation are universal reactions to inflammatory brain injury. We have incorporated this into the Results section 2.4 (line 147) and 2.5 (lines 172, 173).

1-3) The reviewer stated, “The authors have published a series of interesting studies regarding thrombin-preconditioning of MSCs. It seems like the beneficial effects of thrombin preconditioned MSCs is due to the increase production of extracellular vehicles (EVs). How is the transplantation of preconditioned MSCs compared to EVs secreted from the preconditioned MSCs in treating IVH in vivo?”

-> We appreciate the reviewer’s comments. In the present study, because we did not directly compare the efficacy of thrombin-preconditioned MSCs and EVs isolated from thrombin-preconditioned MSCs, it is unknown whether the efficacy of these two conditions would be similar. Further studies are required to clarify this point. (lines 275-279)

1-4) The reviewer stated, “In line 224, the author mentioned about the antioxidant effects of MSCs, it is not clear which data supported this statement.”

-> We measured the antioxidant effect of MSCs by measuring malondialdehyde (MDA) level using the Oxiselect TBARS assay kit (MDA Quantitation) in an in vitro model of thrombin-induced neuronal injury because MDA is one of the most commonly used biomarkers for lipid peroxidation. MSCs significantly reduced MDA levels (Figure 1C). We have incorporated this information into the Results section of the revised manuscript (lines 86-87).

1-5) The reviewer stated, “In Method section, Oxiselect TBARS assay is mentioned. Where is the result?”

-> The Oxiselect TBARS assay kit was used to quantify the level of MDA, a marker of oxidative stress. We have incorporated this information into the Methods section (line 323). The result is presented in panel C of Figure 1.

1-6) The reviewer stated, “The levels of secretory inflammatory cytokines were measured from the cerebrospinal fluid. Is the any histological assessment of these cytokines in the brain tissues?”

-> In our previous study, we measured the levels of inflammatory cytokines in the cerebrospinal fluid (CSF) and brain tissue. And we observed an excellent positive correlation between the cytokine levels in the CSF and those in the brain tissue (References 4 and 26). Thus, in the present study, we measured cytokine levels only in the CSF and not in brain tissue (lines 26-27).

Reviewer 2 Report

The authors analyzed the impact of thrombin preconditioning on the therapeutic potential of mesenchymal stem cells in severe intraventricular hemorrhage, showing that thrombin priming significantly reduces the markers of brain injury as well as functional impairments.

The study is innovative, however, information about the experimental procedures and data analysis, crucial to evaluate the presented results' representativeness, are missing.

Major points:

-All figure legends must indicate the number of animals/group to which the data presented in the graphs correspond. Moreover, it would be more informative about the variability of the data if the graphs were presented as dot plots

Figure 1

-Image presented in panel D should be improved- Due to the small size the figure is of no use since it does not allow you to see the differences expressed in the graph of the same panel.

 - Panel E -The authors must clarify whether BNDF was determined in cell extracts or in the medium. The legend of this panel would be more informative if stating “BDNF protein levels in…”.

-Figure legend - The figure legend should indicate how many independent cell preparations the data shown in the graphs correspond to.

Section “Survival and Body Weight”

- It would be convenient to include an introductory sentence to the studies in the animal model.

- The data presented show that the IVH treatment with naïve MSCs (INM) group has a higher mortality rate than the IVH control (CI) group. These data are somewhat surprising and seem to be out of agreement with the lower cell death, microgliosis, and recovery of the INM group compared to the CI group. The discussion should address these apparently non-concordant data. - In addition, to assess the relevance of the survival data more accurately, the initial and final number of animals per group should be indicated.

Section “Reactive Gliosis and Cell Death”

- The term "light intensity" is rarely used in this context, and it is more common to use the designation "fluorescence intensity".

- The sentence “However, the reduction in the number of TUNEL-positive cells was more significant after transplantation of thrombin-preconditioned MSCs than after transplantation of naïve MSCs” (lines 160-162), is not correct. When compared to the IC group both INM and ITM show significant differences. The authors can refer that the effect observed in the ITM group is higher than that of the INM group (as there are statistically significant differences when comparing these two groups), but not that the effect is more significant.

- The images for panel A (GFAP labeling) are not representative of the data shown in the respective graph since the fluorescence intensity for GFAP is clearly much higher in the IC group compared to the INM group, contrary to what is shown in the graph.

- Legend of figure 3 - Revise the sentence "There are immunofluorescence micrographs of the ventricle area with staining and average graphs for (A) GFAP (red), (B) TUNEL (green), and (C) ED-1 (red). The nucleus was visualized with DAPI (blue) (original magnification, ×200). Each graph is represented by an average."  The parts underlined and highlighted do not seem to make sense.

-The legend should indicate how many animals the data presented in the graphs correspond to, how many slices were analyzed per animal, and how many fields/slice. The information can be provided in the legend or, alternatively, detailed in the methods section.

Brain Inflammation

- The sentence "However, the reduction in the number of TUNEL-positive cells was more significant after transplantation of thrombin-preconditioned cells...", in line 173, is not correct and should be revised. According to the data presented, the INM group does not show any reduction in the inflammatory cytokines analyzed when compared to the IC group. Only the ITM group presents a reduction of cytokines when compared to the IC group.

Section Behavioral Function Test

In Figure 5 the graphs in panels A and B have no legend on the Y-axis. In panel C the units the time unit should be represented as "s" and not "sec".

In panel B it is not clear what differences the asterisk in day 1 refers to. 

Regarding the data presented in panel C, it should be indicated if there are statistically significant differences between group IC and INM.

In the figure legend, the following information should be revised "# p < 0.05 compared with the IVH treatment 207 with thrombin-preconditioned Wharton’s jelly-derived MSC group ". According to what is shown in the graph # indicates the difference in relation to the group itself.

In line 192, it is mentioned “behavioral function” that is a broad term, since the sentence is related to the results from the rotarod test it will be more accurate and precise to correct to “motor function”.

In line 193 should be included in the sentence from which test day are the described results.

In line 195, the beginning of the sentence should be revised since the effect of the preconditioned MSCs was assessed in a separate experimental group (ITM) that was administered with the cells 2 days after the lesion induction, thus this part of the sentence is misleading.

In line 197, when describing the results from the passive avoidance test the most correct parameter to measure is the latency to enter the dark chamber and not “reaction time”.

Materials and Methods

The “4.1. Cell Preparation” section should mention which was the concentration of thrombin used for the preconditioning of the MSCs and the name of the section should be correct to “Mesenchymal Stem Cell Preparation” to better distinguish it from the in vitro model.

The "4.2. In Vitro Model of Thrombin-Induced Neuronal Injury" should be described in more detail. It should be indicated whether cortical cells and MSCs are cultured simultaneously or if MSCs are added later (in which case detail when this was done).

An ethical statement for the approval of the in vivo experiments is missing.

In the “4.4. Animal Model” section (line 327) - instead of referring “slowly injected” it should be specified which was the rate used for the administration (µl/minute or other appropriate units).

Statistical Analysis

In all data presented the statistical analyses have the same p-value (p<0.05). If we take as an example the data presented in figure 3 for the analysis of ED1 labeling, the differences between the NC and CI groups suggest higher significance levels than the differences between the NC and ITM groups, however, what is indicated is that both have a p-value <0.05. This comment applies to all graphs.

Minor po - Typos

Line 23 – “transplantaion” correct to “transplantation”

Line 190 – “the difference was not significant” correct to “the differences were not significant”

Line 221- “which is increases in hemorrhagic” correct to “which is increased in hemorrhagic”

Line 260 – “tropic factors” correct to “trophic factors”

Throughout the text the authors abbreviate mesenchymal stem cells with “MSC” and “MSCs”, this should be corrected to have only one form of abbreviation in all the sections.

Author Response

2-1) The reviewer stated, “All figure legends must indicate the number of animals/group to which the data presented in the graphs correspond. Moreover, it would be more informative about the variability of the data if the graphs were presented as dot plots.”

-> According to the reviewer’s recommendation, we have indicated the number of animals/group in all figure legends and have presented information on the variability of data as dot plots in Figures 1-5 in the revised manuscript.

  • Figure 1

2-2) The reviewer stated, “Image presented in panel D should be improved- Due to the small size the figure is of no use since it does not allow you to see the differences expressed in the graph of the same panel.”

->To show the image in detail, we have enlarged the image in Figure 1D and supplied it as supplementary material (Supplementary Figure S2); we have also incorporated this into the Figure 1D legend (lines 104-105) in the revised manuscript.

2-3) The reviewer stated, “Panel E -The authors must clarify whether BNDF was determined in cell extracts or in the medium. The legend of this panel would be more informative if stating “BDNF protein levels in…”.”

->We determined the BDNF protein levels in the culture medium. According to the reviewer’s recommendation, we have incorporated this into the Figure 1E legend (lines 105-106).

2-4) The reviewer stated, “Figure legend - The figure legend should indicate how many independent cell preparations the data shown in the graphs correspond to.”

-> All in vitro analyses in Figure 1 were performed using MSCs that were prepared at one time. We have incorporated this information into the Figure 1 legend (lines 106-107).

  • Section “Survival and Body Weight”

2-5) The reviewer stated, “It would be convenient to include an introductory sentence to the studies in the animal model.”

-> According to the reviewer’s recommendation, we have included an introductory sentence on the animal model as follows: “NeonatalIVH was induced at postnatal day 4 (P4) by infusing 150 μL of fresh whole blood obtained from the mother rat into the bilateral ventricles respectively. Two days after IVH induction, naïve or thrombin-preconditioned MSCs were transplanted into the ventricle. Survival rate and body weight were observed from P4 to P35.” This introductory sentence on the animal model has been inserted in the “Survival and body weight” section in the Results (lines 114-117).

2-6) The reviewer stated, “The data presented show that the IVH treatment with naïve MSCs (INM) group has a higher mortality rate than the IVH control (CI) group. These data are somewhat surprising and seem to be out of agreement with the lower cell death, microgliosis, and recovery of the INM group compared to the CI group. The discussion should address these apparently non-concordant data.”

-> In the present study, there was no statistical difference in survival rate between the IVH control (IC) group and IVH with naïve MSC treatment (INM) group (7 of 36 died in the IC group and 8 of 31 died in the INM group). However, naïve MSCs showed neuronal cell protection in an in vitro study. Additionally, naïve MSCs significantly attenuated the number of dead cells and activated microglia and slightly improved sensorimotor functions in surviving animals in vivo. Therefore, naïve MSCs may be efficacious, but less potent than thrombin-preconditioned MSCs, in significantly improving mortality rate, anti-inflammation, and sensorimotor function.

2-7) The reviewer stated, “In addition, to assess the relevance of the survival data more accurately, the initial and final number of animals per group should be indicated.”

-> We have indicated the initial and final numbers of animals per group in the legend of Supplementary Figure 1.

  • Section “Reactive Gliosis and Cell Death”

2-8) The reviewer stated, “The term "light intensity" is rarely used in this context, and it is more common to use the designation "fluorescence intensity".”

-> According to the reviewer’s recommendation, we have corrected the term “light intensity” to “fluorescence intensity” in the Results and Method sections and in the Figure 3A legend (lines 148, 150, 372, 373).

2-9) The reviewer stated, “The sentence “However, the reduction in the number of TUNEL-positive cells was more significant after transplantation of thrombin-preconditioned MSCs than after transplantation of naïve MSCs” (lines 160-162), is not correct. When compared to the IC group both INM and ITM show significant differences. The authors can refer that the effect observed in the ITM group is higher than that of the INM group (as there are statistically significant differences when comparing these two groups), but not that the effect is more significant.”

-> According to the reviewer’s recommendation, we have corrected the sentence to “However, the reduction in the number of TUNEL-positive cells was higher in the ITM group than in the INM group” (lines 169-170).

2-10) The reviewer stated, “The images for panel A (GFAP labeling) are not representative of the data shown in the respective graph since the fluorescence intensity for GFAP is clearly much higher in the IC group compared to the INM group, contrary to what is shown in the graph.”

-> According to the reviewer’s recommendation, we have replaced the representative image of the INM group for Figure 3A.

2-11) The reviewer stated, “Legend of figure 3 - Revise the sentence "There are immunofluorescence micrographs of the ventricle area with staining and average graphs for (A) GFAP (red), (B) TUNEL (green), and (C) ED-1 (red). The nucleus was visualized with DAPI (blue) (original magnification, ×200). Each graph is represented by an average." The parts underlined and highlighted do not seem to make sense.”

-> In the legend of Figure 3, we have corrected the sentence to “Immunofluorescence micrographs of the ventricle area with staining and average graphs for (A) GFAP (red), (B) TUNEL (green), and (C) ED-1 (red). The nucleus was visualized with DAPI (blue) (original magnification, ×200)” according to the reviewer’s recommendation (lines 156-158).

2-12) The reviewer stated, “The legend should indicate how many animals the data presented in the graphs correspond to, how many slices were analyzed per animal, and how many fields/slice. The information can be provided in the legend or, alternatively, detailed in the methods section.”

-> The number of animals per group is indicated. In the histological analysis, we observed a total of six fields; two fields (right and left ventricles) were captured in one brain slice, and three brain slices were analyzed. We have incorporated this information into the Methods section (lines 375-377 and lines 387 and 389) and the Figure 3 legend.

  • Brain Inflammation

2-13) The reviewer stated, “The sentence "However, the reduction in the number of TUNEL-positive cells was more significant after transplantation of thrombin-preconditioned cells...", in line 173, is not correct and should be revised. According to the data presented, the INM group does not show any reduction in the inflammatory cytokines analyzed when compared to the IC group. Only the ITM group presents a reduction of cytokines when compared to the IC group.”

-> We have revised the sentence “however, this reduction was more significant after transplantation of thrombin-preconditioned MSCs than after transplantation of naïve MSCs” to “However, the reduction in the number of ED-1-positive cells was higher in the ITM group than in the INM group (lines 176-177).

  • Section Behavioral Function Test

2-14) The reviewer stated, “In Figure 5 the graphs in panels A and B have no legend on the Y-axis. In panel C the units the time unit should be represented as "s" and not "sec".”

-> We apologize for the missing legend on the Y-axis in Figure 5. We have indicated the Y-axis legend and corrected the time units to “s” in Figure 5.

2-15) The reviewer stated, “In panel B it is not clear what differences the asterisk in day 1 refers to.”

-> We performed the rotarod test on 3 consecutive days (from postnatal day (P) 31 to P33). Therefore, the term “day 1, day 2, day3” in the panel B of Figure 5 have been corrected to “postnatal day (P) 31, P32, P33,” in our revised manuscript. Meaningful differences in rotarod performance skill among the groups were observed from P32. Thus, we have presented asterisks at P32 and P33.

2-16) The reviewer stated, “Regarding the data presented in panel C, it should be indicated if there are statistically significant differences between group IC and INM.”

-> We reconfirmed significant differences between the IC and INM groups using a statistical program. However, there was no statistically significant difference between the groups when statistical comparisons between the groups were performed using one-way ANOVA and Tukey’s post hoc analysis.

2-17) The reviewer stated, “In the figure legend, the following information should be revised "# p < 0.05 compared with the IVH treatment with thrombin-preconditioned Wharton’s jelly-derived MSC group ". According to what is shown in the graph # indicates the difference in relation to the group itself.”

-> We apologize for this error. # indicates p < 0.05, compared with the IC (IVH control) group. We have corrected this in the Figure 5 legend.

2-18) The reviewer stated, “In line 192, it is mentioned “behavioral function” that is a broad term, since the sentence is related to the results from the rotarod test it will be more accurate and precise to correct to “motor function”.”

-> According to the reviewer’s recommendation, we have corrected the term “behavioral function” to “motor function” in the revised manuscript (lines 203, 204).

2-19) The reviewer stated, “In line 193 should be included in the sentence from which test day are the described results.”

-> We performed the rotarod test on 3 consecutive days (from P31 to P33). We observed a significant improvement in the ITM group compared with the IC group at P33. We have incorporated this information into the Results section (lines 202 and 206) and the Figure 5 legend.

2-20) The reviewer stated, “In line 195, the beginning of the sentence should be revised since the effect of the preconditioned MSCs was assessed in a separate experimental group (ITM) that was administered with the cells 2 days after the lesion induction, thus this part of the sentence is misleading.”

-> We appreciate the reviewer’s helpful comment. To avoid any misunderstanding, we have revised the sentence to “However, the reduced latency significantly improved in the ITM group compared with that in the IC group” (lines 210-211).

2-21) The reviewer stated, “In line 197, when describing the results from the passive avoidance test the most correct parameter to measure is the latency to enter the dark chamber and not “reaction time”.”

-> We have corrected this parameter to “latency (s)” in the Results section (line 208).

  • Materials and Methods

2-22) The reviewer stated, “The “4.1. Cell Preparation” section should mention which was the concentration of thrombin used for the preconditioning of the MSCs and the name of the section should be correct to “Mesenchymal Stem Cell Preparation” to better distinguish it from the in vitro model.”

-> We have indicated the concentration of thrombin (2 U/mL) used for preconditioning MSCs (line 300). Additionally, the name of the section has been corrected to “Mesenchymal Stem Cell Preparation” according to the reviewer’s recommendation (line 298).

2-23) The reviewer stated, “The "4.2. In Vitro Model of Thrombin-Induced Neuronal Injury" should be described in more detail. It should be indicated whether cortical cells and MSCs are cultured simultaneously or if MSCs are added later (in which case detail when this was done).”

-> Neuronal cells were exposed to thrombin (40 U) overnight to mimic in vivo hemorrhagic conditions. After thrombin-induced neuronal cell injury, naïve or thrombin-preconditioned MSCs were treated with a concentration of 1 × 105 cells/mL (lines 313-315).

2-24) The reviewer stated, “An ethical statement for the approval of the in vivo experiments is missing.”

-> All animal experiments were reviewed and approved by the Institutional Review Board of Sungkyunkwan University. This study followed the institutional and National Institutes of Health guidelines for laboratory animal care. All animal procedures were performed in an AAALAC-accredited specific pathogen-free facility at Sungkyunkwan University. The ethical statement regarding in vivo experiments has been indicated in the Methods section (lines 327-330).

2-25) The reviewer stated, “In the “4.4. Animal Model” section (line 327) - instead of referring “slowly injected” it should be specified which was the rate used for the administration (µl/minute or other appropriate units).”

-> According to the reviewer’s recommendation, we corrected the sentence to “injected at a rate of 80 μL/min with 300 μL of fresh whole blood” (line 337) and to “naïve or thrombin-preconditioned MSCs were injected at a very low rate (10 μL/min) into the right ventricle under stereotaxic guidance” in the revised version of the manuscript (line 347).

  • Statistical Analysis

2-26) The reviewer stated, “In all data presented the statistical analyses have the same p-value (p<0.05). If we take as an example the data presented in figure 3 for the analysis of ED1 labeling, the differences between the NC and IC groups suggest higher significance levels than the differences between the NC and ITM groups, however, what is indicated is that both have a p-value <0.05. This comment applies to all graphs.”

-> According to the reviewer’s recommendation, we have presented the statistical analysis data in detail by indicating p-values when p < 0.01 and p < 0.05 in our revised manuscript.

  • Minor po – Typos

Line 23 – “transplantaion” correct to “transplantation”

Line 190 – “the difference was not significant” correct to “the differences were not significant”

Line 221 – “Which is increases is hemorrhagic” correct to “which is increased in hemorrhagic”

Line 260 – “tropic factors” correct to “trophic factors”

Throughout the text the authors abbreviate mesenchymal stem cells with “MSC” and “MSCs”, this should be corrected to have only one form of abbreviation in all the sections.

-> We thank the reviewer for detailed comments on our manuscript. We have corrected all typographical errors in our revised manuscript (lines 23, 200, 232, and 271).

Round 2

Reviewer 2 Report

The authors made the requested corrections and clarified the issues raised, so I believe that the manuscript can be accepted for publication in its current form.